# Flexural Strength of CAD/CAM Lithium-Based Silicate Glass–Ceramics: A Narrative Review

**DOI:** 10.3390/ma16124398

**Published:** 2023-06-15

**Authors:** Alvaro Munoz, Zejiao Zhao, Gaetano Paolone, Chris Louca, Alessandro Vichi

**Affiliations:** 1Faculty of Dentistry, University of Chile, Santiago 8380544, Chile; 2Dental Academy, University of Portsmouth, Portsmouth PO1 2QG, UK; 3Dental School, IRCCS San Raffaele Hospital, Vita-Salute University, 20132 Milan, Italy

**Keywords:** lithium silicate, lithium disilicate, ceramics, flexural strength, CAD/CAM

## Abstract

Amongst chairside CAD/CAM materials, the use of lithium-based silicate glass–ceramics (LSGC) for indirect restorations has recently been increasing. Flexural strength is one of the most important parameters to consider in the clinical selection of materials. The aim of this paper is to review the flexural strength of LSGC and the methods used to measure it. Methods: The electronic search was completed within PubMed database from 2 June 2011 to 2 June 2022. English-language papers investigating the flexural strength of IPS e.max CAD, Celtra Duo, Suprinity PC, and n!ce CAD/CAM blocks were included in the search strategy. Results: From 211 potential articles, a total of 26 were identified for a comprehensive analysis. Categorization per material was carried out as follows: IPS e.max CAD (n = 27), Suprinity PC (n = 8), Celtra Duo (n = 6), and n!ce (n = 1). The three-point bending test (3-PBT) was used in 18 articles, followed by biaxial flexural test (BFT) in 10 articles, with one of these using the four-point bending test (4-PBT) as well. The most common specimen dimension was 14 × 4 × 1.2 mm (plates) for the 3-PBT and 12 × 1.2 mm (discs) for BFT. The flexural strength values for LSGC materials varied widely between the studies. Significance: As new LSGC materials are launched on the market, clinicians need to be aware of their flexural strength differences, which could influence the clinical performance of restorations.

## 1. Introduction

The use of all-ceramic restorations has steadily increased in recent years, especially since the introduction of CAD/CAM processing techniques [1]. Lithium silicate glass ceramics are popular materials, used since the late 1990s in the field of restorative dentistry due to their good esthetic properties, high strength, superior chemical characteristics, and improved processing capabilities. The original commercial restorative material, IPS Empress 2 from Ivoclar Vivadent, was composed of 65 vol% lithium disilicate in a glass matrix [2]. Improvements in processing parameters allowed for the formation of smaller and uniformly distributed crystals, which translated into superior mechanical characteristics and optical features [3]. This new formulation, called IPS e.max Press, presented tightly packed elongated disilicate crystals, providing crack-bridging characteristics added to a thermal expansion mismatch between filler and glass matrix that causes tangential, compressive stress around the crystals delivering higher strength and toughness to the restorative material [2,3]. These characteristics allowed for IPS e.max Press to be used as a monolithic restoration. These materials have a wide range of indications, such as inlays, onlays, veneers, partial crowns, single crowns in the anterior region and small three-unit FDPs in the anterior and premolar region [4].

CAD/CAM technology has allowed for dentists and technicians to reduce the time required to produce indirect restorations, minimizing inaccuracies, residual stresses, and cross-contamination, thus improving the mechanical performance of the final restoration [5,6]. To utilize these advantages, IPS e.max CAD was launched in 2006. This material comes as a pre-crystallized block, containing 32 vol% of metasilicate (Li_2_SiO_3_) crystals and 0.7 vol% lithium disilicate (Li_2_Si_2_O_5_) crystal nuclei, displaying a flexural strength of around 130 MPa, making the milling process easier. This restoration is later crystallized in a ceramic oven at 850 °C in a vacuum for 20–25 min, where the metasilicate is dissolved and crystallizes as lithium disilicate (~60 vol%), changing from a bluish color to the chosen shade and translucency, and increasing its flexural strength to around 360 MPa [7,8,9,10,11].

Strength can be defined as the ultimate stress a material is able to resist before fracture or plastic deformation and is influenced by the size of flaws and defects on the material surface [12]. However, thermal and mechanical processes can produce microcracks and defects, which also influence strength measurements [13,14]. Flexural strength is a significant mechanical property used to evaluate the strength of brittle ceramic materials, as ceramics are much weaker in tension than in compression [15]. In the oral environment, restorations should have sufficient strength to withstand repeated masticatory forces. Flexural strength could be used to evaluate the maximum force before breakage and can help to predict the performance of the restorations [16,17]. 

Due to the popularity and growing positive reputation of lithium disilicate, other lithium-based silicate glass–ceramics (LSGC) have been marketed in the last few years for their use as chairside CAD/CAM materials [18]. LSGC are composed of Li_2_O, as the main oxide, and SiO_2_. According to the prevalent phase in which materials are crystallized, a classification has been proposed: “lithium disilicate” for those mainly composed of the Li_2_Si_2_O_5_ phase, “lithium silicate” for those crystallizing mainly in the Li_2_SiO_3_ phase, and “lithium (di)silicate” for materials composed of significant fractions of both phases [8]. After an analysis of the CAD/CAM systems became available on the dental market, the following LSGCs were identified: IPS e.max CAD (Ivoclar Vivadent, Schaan, Liechtenstein), Suprinity PC (VITA ZahnFabrik, Bad Sackingen, Germany), Celtra Duo (Dentsply Sirona, Charlotte, NC, USA), GC Initial LiSI Block (GC, Tokyo, Japan), n!ce (Straumann, Basel, Switzerland). The materials and their chemical nature are listed in Table 1.

Some of the CAD/CAM LSGCs (IPS e.max, Suprinity PC), in order to achieve their final mechanical and optical properties, require a thermal treatment (crystallization) that needs to be performed in a dental furnace, while others (Celtra Duo, n!ce, GC Initial LiSI Block) do not require thermal treatment, as they are crystallized by the manufacturer [8]. These latter materials will be increasingly used; therefore, it is important to know whether independent studies have been performed. This review aims to compare the reported flexural strength of LSGC materials and to report on the testing methods used for its determination.

## 2. Materials and Methods

Search strategy

An electronic search of the literature was performed on PubMed on 2 June 2022. The survey covered the period from 2 June 2011 to 2 June 2022. 

The search strategy included the following: ((e.max) OR (Celtra) OR (n!ce) OR (LiSi Block) OR (Suprinity)) AND (strength). 

Inclusion criteria:

Full-text, English-language publications analyzing the strength of CAD/CAM LSGC were included. 

Exclusion criteria:

Publications were excluded if the assessment of strength did not refer to mechanical properties and the investigations did not perform flexural strength testing (3-Point Bending Test; 4-PBT or Biaxial Flexural Test). 

Selection of studies:

After retrieving articles from the electronic search, the titles, followed by the abstracts, were screened according to the inclusion criteria. After screening, full-text articles were assessed and those in compliance with the standards were selected.

Data Extraction:

The following data were extracted from the selected studies using a bespoke data collection form: (1) primary authors; (2) material type; (3) flexural strength test; (4) specimen size; (5) span distance/diameter; (6) crosshead speed; (7) flexural strength value obtained; (8) SEM analysis observation performed; (9) fractography performed. The variables were recorded and tabulated in Excel spreadsheets. The studies in which data on a certain variable were lacking or could not be calculated were entered as “n.r.: not reported” for the variable in question.

## 3. Results

The electronic search identified 211 articles; 182 of them were excluded after reading the titles and abstracts. The 29 remaining articles were analyzed by reading the full text. Finally, 26 articles complied with all the criteria and were included in this review. A flowchart of the study selection process is presented in Figure 1.

Amongst the retrieved articles (Table 2), four LSGC materials were analyzed; IPS e.max CAD (n = 25), Suprinity PC (n = 7), Celtra Duo (n = 5), and n!ce (n = 1). The most common test used to evaluate flexural strength was the 3-PBT (n = 18), followed by BFT (n = 8); only Wendler et al. [19] used BFT and 4-PBT to examine IPS e.max CAD. Most of the studies using BFT tested LSGC discs of 12 mm × 1.2 mm (n = 5), whereas in half of those using 3-PBT, the plate specimens had a dimension of 4 mm × 1.2 mm (width × thickness). There was great variability in the chosen span distance in studies using 3-PBT, with 10 mm and 12 mm used in four studies. In contrast, 7 out of 8 studies using BFT expressly report a 10 mm span distance. Five BFT and ten 3-PBT articles were conducted with a 1 mm/min crosshead speed; the second most common speed was 0.5 mm/min.

Most studies (23 of 26) reported flexural strength average and standard deviation data. Eleven of these also calculated the Weibull modulus and the Weibull characteristic strength. The remaining three studies opted to provide only Weibull characteristic strength.

The reported mean values of flexural strength varied between materials, tests, and within the same materials using the same methods of testing. The largest highest/lowest flexural strength differences were found for Vita Suprinity using BFT in 2 groups (205.89–510 MPa), and IPS e.max CAD using 3-PBT in 20 groups (210.2–471.2 MPa), even though 12 of them ranged between 341 and 398 MPa. At the other end were Celtra Duo and n!ce using BFT, ranging from 177 to 190 MPa and 206 to 222 MPa, respectively. However, these last two materials were tested in the same study comparing two different translucencies: A2LT and A2HT.

## 4. Discussion

In modern dentistry, CAD/CAM ceramics have become very popular restorative materials. Since these materials are required to withstand a hostile oral environment, it is paramount that the mechanical properties of any new ceramics are fully investigated before their clinical use [24]. The property commonly used to classify, compare, and rank materials is strength [19]. Testing this property can shed light on the microstructure, residual stresses, and size, type, and distribution of defects [19]. Flexural strength is a significant property to evaluate the strength of brittle ceramic materials against deformation, as ceramics are much weaker in tension than in compression [15]. It is important to evaluate the maximum force required to fracture a sample of a defined diameter [39]. Flexural strength is usually measured with the 3-PBT, 4-PBT, and BFT; these methods are described in ISO 6872 Dentistry—Ceramic Materials [41]. The present review found that 3-PBT seems to be the preferred testing method within the dental research community.

According to the aforementioned ISO standard, the recommended specimen dimension for the three-point bending test was set at 4 ± 0.2 mm (width), 2.1 ± 1.1 mm (thickness), and 0.12 ± 0.03 mm (chamfer). The specimens must be at least 2 mm longer than the two support rollers (span between 12 ± 0.2 and 40 ± 0.5 mm, and should be loaded at a crosshead speed of 1 ± 0.5 mm/min. Although 15 of the 18 studies used 3-PBT complied with the recommended width, thickness, and speed, only 5 of them explicitly mentioned the presence of a chamfer. In addition, only 11 studies followed the recommended span distance, and four used a 10 mm distance. Variations in the shape and size of the specimens, flaws found on the edges, and set up conditions, all affect flexural strengths [19,42]. Hence, the lack of consistency in testing methodologies found across the articles can help to explain the wide range of 3-PBT values.

For biaxial flexural strength testing, the ISO recommends disc-shaped specimens with a thickness of 1.2 ± 0.2 mm and a diameter of 14 ± 2 mm. The specimens should be placed on three supporting balls 120° apart (creating a circular support of 11 ± 1 mm diameter). A film of non-rigid material is used to evenly distribute contact pressures between the supporting balls and the specimen and another between the loading piston and the specimen. The loading speed is 1 ± 0.5 mm/min. This review found that 7 out of 8 studies used BFT specimens within the recommended dimensions and set-up conditions. However, only Campanelli de Morais et al. [20] and Kang et al. [16] mention the use of load-distributing plastic films. Contrary to the 3-PBT studies, BFT articles showed close similarities between their methods. Nevertheless, a wide spread of flexural data was seen, for example, from six studies reporting IPS e.max CAD values, three obtained flexural strength below 300 MPa [20,33,40], while the other half averaged over 400 MPa [11,16,36]. BFT is considered a solution for small specimens, minimizing the edge effect occurring in uniaxial tests [43], and can simulate real multiaxial stress conditions [19]. Despite these advantages, almost perfectly flat parallel surfaces are required, and errors can be further introduced as friction between the specimen and the supporting fixture during the test [19], which might be responsible for the dissimilar values observed in this review.

Although the values for flexural strength achieved by the glass ceramics showed a wide variation, e.max CAD was stronger than the other three materials, with Celtra DUO and Suprinity obtaining similar results, and n!ce proved to be the weakest material. These results can be explained by the different compositions of the LSGCs. For example, Suprinity and Celtra Duo are mainly composed of lithium oxide and zirconia in a silicon dioxide glassy content, where zirconia is added in at from 8% to 12% in order to increase strength; however, there is a low degree of conversion from lithium metasilicate to lithium disilicate in comparison to e.max CAD [25]. In fact, increasing zirconia content hampers crystal growth [44]. Thus, zirconia lithium silicates also described as lithium (di)silicates (e.g., Suprinity) can be seen by scanning electron microscopy as a fine crystalline structure and e.max CAD as needle-shape crystals of lithium disilicate [6,15,25,26]. In addition, it was possible to observe a slight difference between Suprinity and Celtra Duo, where the latter demonstrated better results. Two possible explanations are the higher residual zirconia in Suprinity (~12 mol% vs. ~14 mol%) and the slight crystallization fraction difference, and the lower vol% of metasilicate in Celtra Duo are due to its being crystallized by the manufacturer [8]. N!ce has been categorized as a lithium aluminosilicate, which is the main crystalline phase with 41 vol% [8], and a much lower lithium disilicate phase in comparison to the other LSGCs mentioned here.

One of the possible limitations of the present review is that only one database has been interrogated. However, as this literature review aimed to provide a reference of practical interest to the dental community, it was decided to formulate the query using commercial names; therefore, a single, widely used database was considered sufficient to identify the relevant papers on the subject.

One of the outcomes of the present study is that most of the materials are marketed using data generated by the manufacturers’ internal studies. It is of primary importance that independent studies are conducted in the early marketing of these materials before they are delivered to the market. A timely independent evaluation of their properties could be of primary importance in avoiding clinical failures.

This review also identified variations in the testing methodologies that were reported, which are the main reason for the large variation in the results. It is therefore advisable that, when the testing methodologies are periodically revised, it would be helpful to narrow the recommended indications.

In addition, the flexural strength values indicated reported in the standards (e.g., ISO 67872:205 [41]), for the different clinical indications should also be improved, since the data reported in such standards do not refer to a specific testing methodology.

A comment could also be made regarding the fractographic analysis that allows for an accurate description of failures based on an understanding of the fracture characteristics, which helps to avoid incorrect statements on the causes of failure [45]. In addition, the analysis can reveal processing or design problems, thus helping with the future improvement of materials. Despite the importance of this, only a small portion of studies applied this valuable tool.

## 5. Conclusions

A large variation in values is reported in the literature for the flexural strength of lithium silicate glass ceramics. While one lithium disilicate, the IPS e.max CAD, has been studied extensively, other materials still require evaluation. The preferred flexural strength test within the dental research community is 3-PBT followed by BFS, but the results vary considerably, mostly due to the large variations in the testing protocols used. These tests should be more precisely described in the standards, and there should be a more precise correlation between the clinically required values and testing methodology. The present review is limited to the flexural strength of LSGC; other physical and mechanical properties (e.g., density, strength, porosity, and hardness) could be the object of future review studies.

## Figures and Tables

**Figure 1 materials-16-04398-f001:**
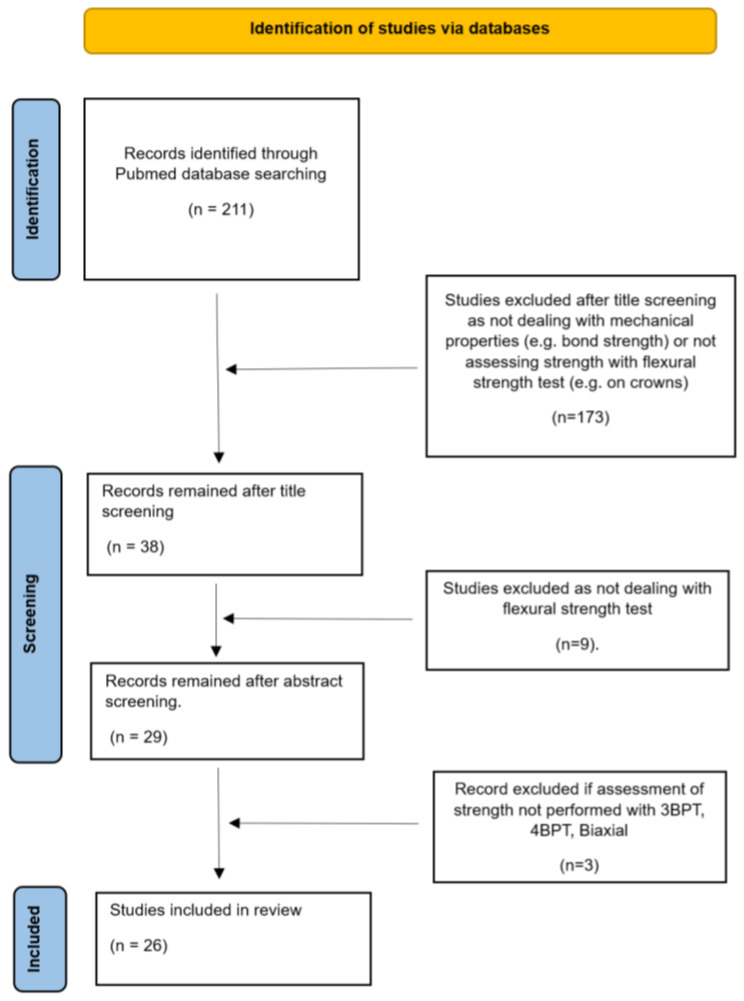
Flowchart of the electronic search and selection of studies.

**Table 1 materials-16-04398-t001:** Chemical composition of the materials included in the review.

Material	Manufacturer	Definition	Chemical Composition	Thermal Treatment
Celtra DUO	Dentsply Sirona, Charlotte, NC, USA	Lithium silicate Zirconia reinforced.	58% SiO_2_; 18.5% Li_2_O; 5% P_2_O_5_; 10.1% ZrO_2_; 1.9% Al_2_O_3_; 2% CeO_2_; 1% Tb_4_O_7_	Optional
Initial LiSi Block	GC Tokyo Japan	Lithium disilicate	55–80% SiO_2_, 10–30% Li_2_O; 5–20% other oxides; pigments: trace	NO
N!ce	Straumann, Basel, Switzerland	Lithium Fluorosilicate	64–70% SiO_2_; 10.5–12.5% Li_2_O; 0–3% K_2_O; 3–8% P_2_O_5_; 0–0.5% ZrO_2_; 10.5–11.5% Al_2_O_3_; 1–2% CaO; 0–9% pigments; 1–3% Na_2_O	NO
IPS e.max CAD	Ivoclar Vivadent Schaan Liechtenstein	Lithium disilicate	57–80% SiO_2_; 11–19% Li_2_O; 0–13% K_2_O; 0–11% P_2_O_5_; 0–8% ZrO_2_, 0–8% ZnO; 0–12% others + colouring oxides	YES
Suprinity PC	VITA Zahnfabrik, Bad Sackingen, Germany	Lithium silicate Zirconia reinforced.	SiO_2_: 56–64% Li_2_O: 15–21% ZrO_2_: 8–12% P_2_O_5_: 3–8% K_2_O: 1–4% Al_2_O_3_: 1–4% CeO_2_: 0–4% Pigments: 0–4%	YES

**Table 2 materials-16-04398-t002:** Characteristics of included studies.

Article	LSGC Type	No. of Specimens	Method	Size (mm)	Span (mm)	Cross-Head Speed (mm/min)	Shade	Translucency	Flexural Strength (MPa)	SD	Characteristic Strength (MPa)	SEM	Fractography
Campanelli de Morais et al. [20]	IPS e.max CAD	15	BFT	12 × 1.2	10	1	n.r.	n.r.	259.39	3.86	n.r.	fx + XRD	yes
	Vita Suprinity	15	BFT	12 × 1.2	10	1	n.r.	n.r.	205.89	2.89	n.r.		
Carrabba et al. [21]	IPS e.max CAD	15	3-PBT	15 × 4 × 1.2	13	1	n.r.	LT	377.00	39	395	qual	no
Elsaka and Elnaghy [15]	IPS e.max CAD	30	3-PBT	18 × 4 × 1.2	16	0.5	n.r.	n.r.	348.33	28.7	361.82	qual	no
	Vita Suprinity	30	3-PBT	18 × 4 × 1.2	16	0.5	n.r.	n.r.	443.63	38.9	406.74		
Fabian Fonzar et al. [22]	IPS e.max CAD	15	3-PBT	16 × 4 × 1.2	13	1	A3	HT	346.21	35.1	359.12	qual	no
	IPS e.max CAD	15	3-PBT	16 × 4 × 1.2	13	1	A3	MT	397.46	62.6	423.39		
	IPS e.max CAD	15	3-PBT	16 × 4 × 1.2	13	1	A3	LT	381.04	42	398.66		
	IPS e.max CAD	15	3-PBT	16 × 4 × 1.2	13	1	A2	MO	281.19	47.9	298.11		
Furtado de Mendonca et al. [6]	IPS e.max CAD	10	3-PBT	14 × 4 × 1.2	10	0.5	n.r.	n.r.	289.00	20	n.r.	qual + EDS	no
	Vita Suprinity	10	3-PBT	14 × 4 × 1.2	10	0.5	n.r.	n.r.	230.00	20	n.r.		
Goujat et al. [23]	IPS e.max CAD	16	3-PBT	18 × 3 × 3	n.r.	0.5	A2	LT	210.20	14	n.r.	no	no
Homaei et al. [24]	IPS e.max CAD	15	3-PBT	14 × 4 × 1.2	12	1	A2	LT	356.70	59.6	381.3	qual + EDS	no
Juntavee and Uasuwan [25]	IPS e.max CAD	15	3-PBT	14 × 4 × 1.2	12	1	A2	HT	378.88	55.4	403.11	qual	yes
	Vita Suprinity	15	3-PBT	14 × 4 × 1.2	12	1	A2	HT	218.41	38.5	234.23		
Kang et al. [16]	IPS e.max CAD	20	BFT	12 × 1.2	10	1	A1	LT	408.30	85.9	n.r.	qual + XRD	no
Kim et al. [26]	IPS e.max CAD	15	3-PBT	17 × 4 × 2	n.r.	0.5	A2	HT	393.43	37.2	409.8	qual + EDS	no
	Celtra Duo	15	3-PBT	17 × 4 × 2	n.r.	0.5	A2	HT	258.92	31.2	272.49		
Kurtulmus-Yilmaz et al. [27]	IPS e.max CAD	10	3-PBT	14 × 4 × 1	10	1	A2	HT	218.10	13.8	n.r.	no	no
Kwon et al. [28]	IPS e.max CAD	10	3-PBT	25 × 4 × 2	20	1	A1	LT	460.00	53	n.r.	no	no
Lawson et al. [29]	IPS e.max CAD	10	3-PBT	16 × 2.5 × 2.5	12	1	A2	LT	376.9	76.2	n.r.	qual + EDS	no
	Celtra Duo	10	3-PBT	16 × 2.5 × 2.5	12	1	A2	LT	300.10	16.8	n.r.		
	Celtra Duo	10	3-PBT	16 × 2.5 × 2.5	12	1	A2	LT	451.40	58.9	n.r.		
Lawson and Maharishi [30]	IPS e.max CAD	10	3-PBT	16 × 4 × 1.2	14	1	BL1	LT	471.22	87.7	n.r.	no	no
Leung et al. [31]	IPS e.max CAD	15	3-PBT	15 × 2 × 2	10	1	A2	LT	341.88	40.3	359.17	qual + EDS	yes
Lien et al. [32]	IPS e.max CAD	12	3-PBT	18 × 4 × 1.3	15	0.5	n.r.	n.r.	362.00	78.6	n.r.	qual + XRD	no
Longhini et al. [33]	IPS e.max CAD	30	BFT	12 × 1.2	10	1	A2	LT	248.60	37.3	257.21	qual	no
Peampring and Sanohkan [34]	IPS e.max CAD	10	3-PBT	20 × 4 × 2	n.r.	0.5	n.r.	n.r.	349.96	38.3	411.52	no	no
Riquieri et al. [35]	Vita Suprinity	30	BFT	12 × 1.2	10	1	n.r.	n.r.	n.r.	n.r.	Unfired: 106.95 (100.94–113.33)	qual + EDS + XRD	yes
	Vita Suprinity	30	BFT	12 × 1.2	10	1	n.r.	n.r.	n.r.	n.r.	Fired: 191.02 (178.10–204.89)		
	Celtra Duo	30	BFT	12 × 1.2	10	1	n.r.	n.r.	n.r.	n.r.	Unfired: 163.86 (153.21–175.26)		
	Celtra Duo	30	BFT	12 × 1.2	10	1	n.r.	n.r.	n.r.	n.r.	Fired: 251.25 (235.36–268.21)		
Şen and Us [36]	IPS e.max CAD	30	BFT	12 × 1.2	10	0.5	A2	HT	415.00	26	429	qual + EDS	no
	Vita Suprinity	30	BFT	12 × 1.2	10	0.5	2M2	HT	510.00	43	532		
Sonmez et al. [37]	IPS e.max CAD	10	3-PBT	14 × 4 × 1.2	12	0.5	A2	HT	359.20	4.2	n.r.	qual + XRD + EDS	no
Stawarczyk et al. [38]	IPS e.max CAD	30	3-PBT	15 × 4 × 3	10	1	A2	HT	n.r.	n.r.	356 (337–374)	no	no
Stawarczyk et al. [11]	IPS e.max CAD	10	BFT	12 × 0.95	10	1	A2	LT	432.00	48.7	452	no	no
	IPS e.max CAD	10	BFT	12 × 0.95	10	1	A2	HT	418.00	57,6	442		
	Celtra Duo	10	BFT	12 × 0.95	10	1	A2	LT	190.00	23.3	200		
	Celtra Duo	10	BFT	12 × 0.95	10	1	A2	HT	177.00	29.1	189		
	N!ce	10	BFT	12 × 0.95	10	1	A2	LT	222.00	28.4	234		
	N!ce	10	BFT	12 × 0.95			A2	HT	206.00	27.7	220		
Tavares et al. [39]	IPS e.max CAD	10	3-PBT	20 × 4 × 1.2	16	0.5	A2	HT	341.45	61.4	n.r.	qual + XRD	no
Wang et al. [40]	IPS e.max CAD	10	BFT	13 × 1.2	10	0.5	A2	HT	295.80	48.9	n.r.	qual + XRD	no
	IPS e.max CAD	10	BFT	13 × 1.2	10	0.5	A2	MO	248.50	30	n.r.		
Wendler et al. [19]	IPS e.max CAD	30	BFT	Disc: 12 × 1.2	n.r.	0.5–1.5	n.r.	n.r.	n.r.	n.r.	647.98 (635.69–660.65)	weakest specimen/group	weakest specimen/group
	IPS e.max CAD	30	BFT	Plate: 12 × 12 × 1.2	n.r.	0.5–1.5	n.r.	n.r.	n.r.	n.r.	609.80 (594.78–625.38)		
	IPS e.max CAD	30	4-PBT	25 × 2.5 × 2	Outer:20 inner:10	0.1	n.r.	n.r.	n.r.	n.r.	462.06 (446.85–477.96)		
	Vita Suprinity	30	BFT	Disc: 12 × 1.2	n.r.	0.5–1.5	n.r.	n.r.	n.r.	n.r.	611.24 (573.80–651.58)		
	Vita Suprinity	30	BFT	Plate: 12 × 12 × 1.2	n.r.	0.5–1.5	n.r.	n.r.	n.r.	n.r.	537.03 (503.77–651.58)		
	Celtra Duo	30	BFT	Disc: 12 × 1.2	n.r.	0.5–1.5	n.r.	n.r.	n.r.	n.r.	626.84 (587.74–669.02)		
	Celtra Duo	30	BFT	Plate: 12 × 12 × 1.2	n.r.	0.5–1.5	n.r.	n.r.	n.r.	n.r.	565.80 (534.02–599.86)		

SD, standard deviation; BFT, biaxial flexural strength test; 3-PBT, three-point bending test; 4-PBT, four point bending test; n.r. not reported; SEM, scanning electron microscopy; qual, qualitative description; XRD, X-ray diffraction; EDS, energy-dispersive spectroscopy.

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
