# Peer review of "Flexural Strength of CAD/CAM Lithium-Based Silicate Glass–Ceramics: A Narrative Review"

_materials, 2023, doi:10.3390/ma16124398_

Round 1
Reviewer 1 Report
This study is important for a dentist community. I advice to transfer the paper to Dental Materials or to a similar journal. In the present form, its content is too specific.
P. 2, line 82. “An electronic search of the literature was performed on PubMed on June 2nd, 2021. The survey covered the period from June 2nd, 2011 to June 2nd, 2022.” Did you mean “An electronic search of the literature was performed on PubMed on June 2nd, 2022.”?
Author Response
This study is important for a dentist community. I advice to transfer the paper to Dental Materials or to a similar journal. In the present form, its content is too specific.
The paper was submitted to the Journal Materials, Section Biomaterials, Special Issue “Materials for Prosthodontics, Restorative, and Digital Dentistry”
The subject of the present papers falls within the scopes of the Special Issue…
- 2, line 82. “An electronic search of the literature was performed on PubMed on June 2nd, 2021. The survey covered the period from June 2nd, 2011 to June 2nd, 2022.” Did you mean “An electronic search of the literature was performed on PubMed on June 2nd, 2022.”?
We apologise for the mistake, and we thank the reviewer for identifying it. When the initial search was updated of one year, we wrongly left one of the original dates, sorry for this inaccuracy.
English has been extensively revised by a mother tongue author.
Reviewer 2 Report
The paper is devoted to a review of the situation with the estimation of the flexural strength of dental glass-ceramic materials based on lithium silicates.
The authors reviewed the publications in the PubMed database and evaluated the mechanical properties of the materials presented in the publications, compared the prevalence of testing techniques aimed at assessing the flexural strength, and formulated the reasons for the difficulty in comparing the results of determining the mechanical properties of the materials described in the selected publications.
The subject of the work is of practical interest to the dental community and can be published after the following refinements:
1. The paper only searched the PubMed database. Why only PubMed? It seems to me that 26 results do not reflect the real picture in the field, I suggest searching additional databases. For example, Scopus returns 601 results for a similar query.
2. Provide Figure 1 in better quality, the text is hard to read.
3. As far as I can tell, your query didn't include Suprinity PC, but it still shows up in the results, and it's the second most common. Maybe you should change your search strategy?
4. The Introduction section should be expanded to include a description of the selected LSGC materials, including numerical values describing the prevalence of the selected materials in the industry, to better assess the significance of the work performed.
5. Please indicate what histologic studies of the tissues will be required after implantation. in vivo on animals?
6. How will your material affect cytological studies, what cultures grow in the mouth?
10.1016/j.ceramint.2019.04.081
7. is morphometry planned for future studies?
10.1016/j.powtec.2020.04.040
8. Is there a body response to materials similar to lithium silicate used in human, animals?
10.1016/j.pnsc.2019.07.004
Author Response
The authors would like to thank the reviewer for his/her work on the paper and suggestions.
The paper is devoted to a review of the situation with the estimation of the flexural strength of dental glass-ceramic materials based on lithium silicates.
The authors reviewed the publications in the PubMed database and evaluated the mechanical properties of the materials presented in the publications, compared the prevalence of testing techniques aimed at assessing the flexural strength, and formulated the reasons for the difficulty in comparing the results of determining the mechanical properties of the materials described in the selected publications.
The subject of the work is of practical interest to the dental community and can be published after the following refinements:
- The paper only searched the PubMed database. Why only PubMed? It seems to me that 26 results do not reflect the real picture in the field, I suggest searching additional databases. For example, Scopus returns 601 results for a similar query.
As the reviewer correctly identified, the aim of the paper was to produce a reference of practical interest to the dental community. For this reason, we decided to formulate the query using commercial names. Even if we agree that the use of several databases could be more formally correct, we considered that if commercial names are used, it is unlike that the use of PubMed will not be sufficient to identify the relevant paper on the subject.
We also kindly ask the reviewer to consider that 26 results are after the screening, the return of the query was 211.
However, on the basis of the comment of the reviewer, we added the use of only one database as a possible limitation of the study.
- Provide Figure 1 in better quality, the text is hard to read.
Figure 1 has been completely redesigned.
- As far as I can tell, your query didn't include Suprinity PC, but it still shows up in the results, and it's the second most common. Maybe you should change your search strategy.
The reviewer is absolutely right, we apologise for the inaccuracy, and we thank the reviewer for identifying it. Suprinity PC was added at a later stage, when the review was lastly updated, but the description of the query was not updated accordingly, sorry for this.
- The Introduction section should be expanded to include a description of the selected LSGC materials, including numerical values describing the prevalence of the selected materials in the industry, to better assess the significance of the work performed.
The introduction section has been expanded to include a description of the selected LSGC materials.
As well, a Table reporting the composition has been added.
As reported in the introduction, these materials represent an important category of prosthetic materials, routinely used in a large scale in dentistry. However, numerical values describing the prevalence of the selected materials in the industry to the knowledge of the authors are not available in scientific literature, rather only in marketing analysis.
Extensive English revision has been performed by a mother tongue author.
. Please indicate what histologic studies of the tissues will be required after implantation. in vivo on animals?
- How will your material affect cytological studies, what cultures grow in the mouth? 10.1016/j.ceramint.2019.04.081
- is morphometry planned for future studies? 10.1016/j.powtec.2020.04.040
- Is there a body response to materials similar to lithium silicate used in human, animals? 10.1016/j.pnsc.2019.07.004
The last three questions of the reviewer are not clear to the authors, as it seems that they are not related with the object of the present study.
Reviewer 3 Report
Your work is interesting, but revisions are recommended, as follows.
1. I am not sure if this review brings new and useful data comparatively to other reviews in domain. Please comment this.
2. The search strategy subchapter is unusual, from my point of view. I don’t know if it is important for the review to be inserted or it would be better to exclude this and include only the review itself.
3. Perhaps some structural data is indicated to be inserted in this review, as the mechanical properties are strongly related to the structure of each sample.
Author Response
Comments and Suggestions for Authors
Your work is interesting, but revisions are recommended, as follows.
- I am not sure if this review brings new and useful data comparatively to other reviews in domain. Please comment this.
To the knowledge of the authors, while there are several papers experimentally testing this relevant property (flexural strength) of one of the most used prosthetic dental materials (lithium disilicate), a paper reporting the results of the various papers and critically comparing them is missing. Therefore, the aim of this study was to review the publications available, comparing the prevalence of testing techniques aimed at assessing the flexural strength, and formulating the reasons for the difficulty in comparing the results.
We think that the paper is of practical interest to the dental community.
- The search strategy subchapter is unusual, from my point of view. I don’t know if it is important for the review to be inserted or it would be better to exclude this and include only the review itself.
We generally report the search strategy as a subchapter in review papers. However, taking into account the comment of the reviewer, the table was removed, and search strategy was reported in the text.
- Perhaps some structural data is indicated to be inserted in this review, as the mechanical properties are strongly related to the structure of each sample.
Some more information about the materials, as well as a Table with the chemical composition, have been added to the paper.
English has been revised by a mother tongue author.
Round 2
Reviewer 1 Report
Before publishing, please correct typos, etc.
Please use subscripts for figures while writing the chemical formulas throughout the manuscript.
Table 2. Please correct typo in the chemical compostion of the Celtra DUO material. It should be written ZrO2 istead of ZrO.
Line 106. Please correct typo in the words “3-point bending test”.
Figure 1 is not readable. Please correct it.
Author Response
Please use subscripts for figures while writing the chemical formulas throughout the manuscript.
We are sorry but the request is not clear to the authors …
There is only 1 figure and it does not contains chemical formulas... maybe the reviewer means something like ZrO2 (with the number 2 in subscript) instead of ZrO2 ?
Can the reviewer please further clarify ?
Thank you
Table 2. Please correct typo in the chemical compostion of the Celtra DUO material. It should be written ZrO2 istead of ZrO.
Right.
Sorry for the typo and thank you.
Line 106. Please correct typo in the words “3-point bending test”.
Right.
Sorry for the typo and thank you again.
Figure 1 is not readable. Please correct it.
MDPI requests to resubmit the manuscript containing the changes made (revision mode).
Even if it was still in the paper as track changes, the figure of line 131-134 has already been replaced with a new figure that is in the same format as previously published several times in MDPI journals.
We hope now it can read well.
Reviewer 2 Report
1.Thank you very much for your comments, but I still have one serious question, where do you place our Lithium-based Silicate Glass-Ceramics applications in biomedicine and dentistry. It looks very strange - please give more detail on this application, or remove the reference to it in general.
lime 252 "The preferred flexural strength test within the dental research community"
2. Again, the objectives set do not correlate in any way with the conclusions.
3. I ask the authors for more details on the subject matter of the journal https://www.mdpi.com/journal/materials/about
"Materials provides a forum for publishing papers which advance the in-depth understanding of the relationship between structure, properties, and functions of all kinds of materials. It covers all aspects of materials science and engineering including synthesis, structure, mechanical, chemical, electronic, magnetic, and optical properties, as well as their various applications. In addition to regular issues, Materials publishes several Special Issues per year on specific topics."
In your study I did not see - "composition-structure properties" only the process itself, I ask for a more detailed emphasis on the fundamental and applied aspects of the use of silicates
4. Authors do not cite MDPI journals please increase the number of citations
5. Line 131 134 drawings are very hard to read - who are the authors writing this article for? The images are very poorly legible.
6. Please explain what exactly to get veneers and present the 3D model. If for this type of products. why only bending strength is considered, when it is very necessary to study all physical and mechanical properties (density, strength, porosity and hardness)
Author Response
Comments and Suggestions for Authors
1.Thank you very much for your comments, but I still have one serious question, where do you place our Lithium-based Silicate Glass-Ceramics applications in biomedicine and dentistry. It looks very strange - please give more detail on this application, or remove the reference to it in general.
LSGC are routinely widely used in dentistry for permanent restorations like inlays, onlays, overlays, structure for single crowns, monolithic single crowns, and with limited indications for structures and/or monolithic for 3-unit bridges, both tooth- and implant- supported, due to a good combination of resistance and aesthetics.
lime 252 "The preferred flexural strength test within the dental research community"
Sentence is incompletely reported.
It is “The present review found that 3-PBT seems to be the preferred testing method within the dental research community”.
- Again, the objectives set do not correlate in any way with the conclusions.
As reported in the text, these were the objectives:
- to compare the reported flexural strength of LSGC materials
and
- to report about the testing methods used for its determination”.
And these are the conclusions:
(i) A large variation of values is reported in the literature for the flexural strength of Lithium Silicate Glass Ceramics. While one lithium disilicate, the IPS e.max CAD, has been studied extensively, other materials still require evaluation.
(ii) The preferred flexural strength test within the dental research community is 3-PBT followed by BFS, but the results reported vary considerably, mostly due to the large variation in the testing protocols used. These tests should be more precisely and more narrowly described in the standards…
Therefore, the objectives and the conclusions does correlate…
- I ask the authors for more details on the subject matter of the journal https://www.mdpi.com/journal/materials/about
"Materials provides a forum for publishing papers which advance the in-depth understanding of the relationship between structure, properties, and functions of all kinds of materials. It covers all aspects of materials science and engineering including synthesis, structure, mechanical, chemical, electronic, magnetic, and optical properties, as well as their various applications. In addition to regular issues, Materials publishes several Special Issues per year on specific topics."
In your study I did not see - "composition-structure properties" only the process itself, I ask for a more detailed emphasis on the fundamental and applied aspects of the use of silicates
The paper was submitted to the Journal Materials, Section Biomaterials, Special Issue “Materials for Prosthodontics, Restorative, and Digital Dentistry”.
This is the information concerning this Special Issue:
...
Considering the current revolution in the development of new materials in combination with the concept of digital dentistry, the aim of this Special Issue entitled “Materials for prosthodontics, restorative dentistry and digital dentistry” is to address the last innovations in the field for material science.
This issue is open to studies that investigate the influence of the material characteristics, behavior and/or the manufacturing process of medical and dental devices through partial or total digital workflows. This Special Issue includes research into materials (in their characteristics, design, manufacturing, and clinical performance phases) for tissue substitutes, dental appliances, or prosthetic supplies. The study of materials science in restorative dentistry, dental and maxillofacial prosthetics, and digital dentistry is contemplated, e.g., fixed and removable devices, complete dentures, partial dentures, splints, dental implants, surgical guides, epitheses, auxiliary devices for preclinical and clinical application, as well as orthopedic and/or orthodontic appliances. Among others, both additive and subtractive manufacturing processes are within the scope of this Special Issue, covering metal alloys, dental ceramics, polymers, composites, and hybrid materials.
…
Therefore, a review on the most important mechanical property of a category of dental ceramics currently largely used in dentistry, with a CAD-CAM digital workflow (subtractive manufacturing) falls within the scopes of the Special Issue.
- Authors do not cite MDPI journals please increase the number of citations
It is a good scientific conduct that papers should be quoted on the basis of their content, and not on the basis of the Publisher.
- Line 131 134 drawings are very hard to read - who are the authors writing this article for? The images are very poorly legible.
MDPI requests to resubmit the manuscript in revision mode.
The figure of line 131-134, if correctly opened in Word, has a red line on it, that means that it has been deleted.
In the present version the track changes of round 1 have been removed, thus we hope now it is clearer which is the correct figure, that is with the same structure we used in several MDPI papers.
Regarding the sentence “who are the authors writing this article for ?” we kindly ask the reviewer to use the same respectful approach to the authors that the authors have in her/his regard.
- Please explain what exactly to get veneers and present the 3D model. If for this type of products. why only bending strength is considered, when it is very necessary to study all physical and mechanical properties (density, strength, porosity and hardness).
The question posed is unclear, as neither veneers nor 3D model have been reported in the paper.
Once again, the paper is a literature review about one relevant mechanical property of a widely used category of dental materials. Of course, there are other important mechanical properties, but they cannot be all reported in one literature review.
However, as we agree with the reviewer on the importance of performing other literature reviews like the present one on other mechanical properties, a sentence in this regard has been added to the discussion.
Reviewer 3 Report
Dear Authors,
You made the comments and revisions which were recommended. I agree the paper to be published din this revised form.
Author Response
Dear Authors,
You made the comments and revisions which were recommended. I agree the paper to be published din this revised form.
Thank you for your time.
Round 3
Reviewer 1 Report
In my previous report, I really meant something like ZrO2 (with the number 2 in subscript) instead of ZrO2 while discussing correction in the column "Chemical composition"of Table 1. This correction can be made during the publishing process.
Reviewer 2 Report
-